# Vaccines against the original strain of SARS-CoV-2 provide T cell memory to the B.1.1.529 variant

Charlyn Dörnte [1], Verena Traska[1], Nicole Jansen[1], Julia Kostyra[1], Herrad Baurmann[1], Gereon Lauer[1], Yi-Ju Huang[1], Sven Kramer[1], Olaf Brauns[1], Holger Winkels [2], Jürgen Schmitz[1], Christian Dose[1], Anne Richter[1] & Marc Schuster [1✉]

## Abstract

**Background** The SARS-CoV-2 variant B.1.1.529 potentially escapes immunity from vaccination via a heavily mutated Spike protein. Here, we analyzed whether T cell memory towards the B.1.1.529 Spike protein is present in individuals who received two or three doses of vaccines designed against the original Wuhan strain of SARS-CoV-2.

**Methods** PBMCs were isolated from two- and three-times vaccinated study participants and incubated in vitro with peptide pools of the Spike protein derived from sequences of the original Wuhan or the B.1.1.529 strains of SARS-CoV-2. Activated antigen-specific T cells were detected by flow cytometry. In silico analyses with NetMHCpan and NetMHCIIpan were used to determine differences in MHC class presentation between the original strain and the B.1.1.529 strain for the most common MHCs in the European-Caucasian population.

**Results** Here we show, that both CD4 and CD8 responses to the B.1.1.529 Spike protein are marginally reduced compared to the ancestor protein and a robust T cell response is maintained. Epitope analyses reveal minor differences between the two SARS-CoV-2 strains in terms of MHC class presentations for the MHC-alleles being most common in the European-Caucasian population.

**Conclusions** The memory T cell response induced via first generation vaccination remains robust and is mostly unaffected by B.1.1.529 mutations. Correspondingly, in silico analyses of MHC presentation of epitopes derived from the B.1.1.529 Spike protein shows marginal differences compared to the ancestral SARS-CoV-2 strain.

## Plain language summary

Vaccination against SARS-CoV-2 results in the production of proteins called antibodies, that bind and inactivate the virus, and cells that help to eliminate it from the body in a future encounter, such as memory T cells. Both antibodies and memory T cells remain in the body after vaccination with memory T cells being present for longer than antibodies. Here, we determined that even though most of the first generation vaccines were created to prevent infection with the original SARS-CoV-2 virus, the memory T cells generated by this vaccination can also detect the omicron variant.

[1] Miltenyi Biotec B.V. & Co. KG, Friedrich-Ebert-Straße 68, 51429 Bergisch Gladbach, Germany. [2] Faculty of Medicine and University Hospital Cologne, Clinic III for Internal Medicine, University of Cologne, Cologne, Germany. ✉email: marcsch@miltenyi.com

The recently emerged SARS-CoV-2 variant B.1.1.529 (Omicron) replaced B.1.617.2 (Delta) as the dominant strain of the COVID-19 pandemic in 2022. Compared to the ancestral strain about 50% of B.1.1.529 mutations are harbored within the Spike protein[1], and as the ancestral sequence was used for developing the majority of first generation vaccines[2], the evaluation of potential immune escape is essential, also for potentially upcoming vaccines and future virus strains. Indeed, limited neutralization of B.1.1.529 virus variants by vaccine-induced antibodies has already been reported[3]. The corresponding epitopes for these neutralizing antibodies are usually restricted to a relatively small region of the total Spike protein sequence, while T cell epitopes are present throughout the whole protein and made up of longer sequences of amino acids[4], suggesting that total T cell responses would be less affected by mutations. According to in silico analyses, among reported SARS-CoV-2 T cell epitopes, 14% of CD8 and 28% of CD4 epitopes harbor at least one mutation[5], which should not strongly abrogate MHC binding or restriction.

In summary, we show here that the T cell responses to the B.1.1.529 Spike protein remain robust in a cohort of study subjects receiving first generation vaccines despite numerous spike protein mutations present in B.1.1.529. In silico prediction of MHC presented B.1.1.529 Spike protein epitopes reveals little differences when compared with the original SARS-CoV-2 Spike protein epitopes, corresponding to the largely unaltered in vitro T cell responses.

## Methods

**Ethical evaluation of the study**. The study was approved as a biomedical research project by the ethical committee of the medical association of North Rhine (ID151/2020).

**Whole blood donations & study participants**. Whole blood donations of 18 healthy volunteers were drawn after informed consent and following the WMA Declaration of Helsinki regarding the ethical principles for medical research involving human subjects. Therefore 30 mL whole blood was drawn using Lithium-Heparin coated blood collection tubes (BD, New Jersey, USA, Cat. No: 367526). All recruited participants had a diverse background regarding vaccination against Covid-19 (Supplementary Table 1).

**PBMC isolation and stimulation with SARS-CoV-2 derived peptides**. To assess the reactivity of SARS-CoV-2 specific T cells, PBMCs from all whole blood samples were isolated via density-gradient centrifugation using Pancoll® (Pan Biotech, Aidenbach, Germany, Cat. No. P04-60500), according to the manufacturer's protocol using CliniMACS® PBS/EDTA Buffer (Miltenyi Biotec, Bergisch Gladbach, Germany, Cat. No. 200-070-025). To remove remaining thrombocytes from isolated PBMCs, samples were washed twice by resuspending them in 50 mL CliniMACS® PBS/EDTA buffer and subsequently centrifuged at 200 g for 15 min. Afterwards, cell numbers were determined using a Sysmex XP-300 device (Sysmex, Norderstedt, Germany). Cells were then plated out in a 96-well flat-bottom plate (Falcon, New York, USA, Cat. No. 353072) at a concentration of $1 \times 10^6$ cells/0.1 mL RPMI-1640 Medium (Biowest, Nuaillé, France, Cat. No. L0501-500) supplemented with 5% human AB Serum (Capricorn, Ebsdorfergrund, Germany, Cat. No. HUM-3B, Lot. CP20-3472) and 1x Gibco Anti-Anti (Thermo Fisher Scientific, Waltham, USA, Cat. No. 11580486) /well. Next, SARS-CoV-2 reactive T-lymphocytes were stimulated by adding 1 µg/mL SARS-CoV-2 derived peptides from A) a pool of 83 15mer-peptides covering all mutations of the B.1.1.529 strain ("B1.1.529 Mutation Pool", Miltenyi Biotec,

Cat. No. 130-129-928), B) a pool of 83 15mer-, reference-peptides to A derived from the wildtype virus ("WT Reference Pool", Miltenyi Biotec, Cat. No. 130-129-927), or C) a mega-pool of 360 15mer-peptides covering the complete sequence of the wildtype Spike protein ("Prot_S Complete", Miltenyi Biotec, Cat. No. 130-127-951). An unstimulated control sample was prepared as negative control. The cells were then incubated for 6 h at 37 °C, 5% CO2. After 2 h of incubation, 2 µg/mL Brefeldin A (Sigma-Aldrich, St. Louis, USA, Cat. No. B7651) was added to each well.

**Staining of activity markers and intracellular cytokines**. Reactivities of CD4 and CD8 T cell subsets after stimulation with SARS-CoV-2 derived peptides were quantified by the staining of the activation-associated marker CD154, together with staining of intracellular cytokines IFN-γ, TNF-α, and IL-2. This staining was performed in a 96-well V-bottom plate (Sigma-Aldrich, St. Louis, USA, Cat. No. Z667234) into which cells were transferred after adding 100 µl PBS/EDTA (2 mM) buffer to each sample. After centrifugation at 300 g for 5 min the supernatant was discarded and dead cells were stained with Viobility™ 450/452 Fixable Dyes (Miltenyi Biotec, Cat. 130-109-816) according to the manufacturer's instructions. Afterward, cells were washed with PBS, centrifuged at 300 g for 5 min, and the supernatant was discarded, followed by the fixation of cells using Inside Fix (Inside Stain Kit, Miltenyi Biotec, Cat. No. 130-090-477), according to the manufacturer's instruction. Subsequently, cells were permeabilized by resuspending and centrifuging them in 100 µl Inside Perm (Inside Stain Kit, Miltenyi Biotec, Cat.-No. 130-090-477), using the described settings. Finally, cells were stained using the following antibody-cocktail: anti-CD3 – APC (Miltenyi Biotec, Cat. No. 130-113-135), anti-CD14 – VioBlue (Miltenyi Biotec, Cat. No.130-110-525), anti-CD20 – VioBlue (Miltenyi Biotec, Cat. No.130-111-531), anti-CD4 – VioBright515 (Miltenyi Biotec, Cat. No.130-114-535), anti-CD8 – VioGreen (Miltenyi Biotec, Cat. No.130-110-684), anti-IFN-γ – PE (Miltenyi Biotec, Cat. No.130-113-496), anti-TNF-α – PEVio770 (Miltenyi Biotec, Cat. No.130-120-492), anti-CD154 – APCVio770 (Miltenyi Biotec, Cat. No.130-114-130), and anti-IL-2 – PEVio615 (Miltenyi Biotec, Cat. No.130-111-307). All antibodies were used in a 1:50 dilution. Staining was done according to the manufacturer's instructions. Cells were washed by addition of Inside Perm and centrifugation at 300 g for 5 min. After discarding the supernatant cells were resuspended in PBS/EDTA/BSA – buffer for subsequent flow-cytometric analysis. Data acquisition was done using MACS-Quant16 – flow cytometer (Miltenyi Biotec, Cat. No. 130-109-803).

**NetMHCpan analysis**. For the prediction and comparison of wildtype and B.1.1.529 derived peptide-binding affinities to MHC class-I and MHC class-II molecules, the open-source software NetMHCpan was used. Therefore, the NetMHCpan-algorithm (https://services.healthtech.dtu.dk/service.php?NetMHCpan-4.1) calculated affinities against MHC Class I molecules. Whereas the NetMHCIIpan-algorithm was used for MHC class II-directed predictions (https://services.healthtech.dtu.dk/service.php?NetMHCIIpan-4.0). As working-output, the EL-Rank was utilized to compare binding affinities of corresponding peptides derived from the wildtype- and B.1.1.529-strain. With these values, we first evaluated the single peptide binding affinities to categorize them into a group of strong binders (SB, EL-Rank < 0.5), weak binders (WB, EL-Rank >0.5, <2), or no-binders (NB, EL-Rank >2) for MHC class I molecules. As peptides binding to MHC class II molecules extrude out of the binding pocket, 9mer-core peptides from the 15mer templates were calculated for each peptide MHC class II combination in order to determine the best binder. For the resulting EL-Rank and aimed categorization of peptides binding to

MHC class II molecules into strong-, weak-, or no-binders previously mentioned EL-Rank-thresholds were adapted according to the software recommendation, to <1; >1, <5; and >5, respectively. With this categorization at hand, the binding evolution going from the wildtype to the B.1.1.529 strain was analyzed, which allowed for the quantitation of switches between every possible category, going from the wildtype virus to the B.1.1.529-variant.

**Statistics and reproducibility**. Flow cytometric data was analyzed using FlowJo 10.7.2. Descriptive statistics presented in this article was conducted using GraphPad Prism version 8.0.0 for Windows (GraphPad Software, San Diego, California USA, www. graphpad.com). The described experimental results were based on the two cohorts "2x vaccinated individuals" and "3x vaccinated individuals", comprising of eight and ten study subjects, respectively. For each stimulation experiment, an unstimulated control was included to assess the individual background signal. In order to judge on inter- and intra-cohort differences, it was first checked whether the data were normally distributed or not, using the GraphPad Prism function "normality and lognormality tests", which uses the Anderson-Darling, D'Agostino & Pearson, Shapiro-Wilks and Kolmogorov-Smirnov tests. These tests revealed that data on $CD154 + IFN\text{-}\gamma + CD4 + T$ cells were normally distributed, whilst data on $IFN\text{-}\gamma + TNF\text{-}\alpha + CD8 + T$ cells were not. On this basis, observed differences between $CD154 + IFN\text{-}\gamma + CD4 + T$ cells stimulated with the described peptide pools, were evaluated using unpaired T-test with Welch's correction. Within the group of $IFN\text{-}\gamma + TNF\text{-}\alpha + CD8 + T$ cells a Mann–Whitney test was used to evaluate potential significances. For the determination of the statistical power, a power analysis using JMP® (Version 15.2.0, SAS Institute Inc., Cary, NC, 1989–2022) was performed. For this analysis, all eight means calculated from the different stimulatory conditions, as well as the greatest standard deviation found for the respective cell type were used. The significance threshold α was set to 0.05.

**Reporting summary**. Further information on research design is available in the Nature Research Reporting Summary linked to this article.

## Results and discussion
We compared the T cell responses upon stimulation with three different Spike protein derived peptide pools in double- and triple-vaccinated study subjects and one subject infected with Covid-19 (Supplementary Table 1). One pool covered the entire ancestral Spike protein (Prot_S Complete), the second pool contained peptides harboring B.1.1.529 (variant BA.1) mutations (Prot_S B.1.1.529 Mutation Pool), while the third pool contained the unmutated reference peptides, relative to the mutation pool (Prot_S WT Reference Pool) (Supplementary Fig. 1). Of note, double- and triple-vaccinated study subjects were not the same individuals and not MHC-matched (Supplementary Table 1). These limitations need to be taken into account concerning all described differences between double- and triple-vaccinated individuals.

Based on CD154, TNF-α, and IFN-γ flow analysis, and calculated frequencies for reactive CD4 or CD8 T cells, we determined that the Prot_S Complete pool stimulated significant CD4 and CD8 responses after vaccination, as reported before (Fig. 1a, b, Supplementary Data 1, Supplementary Fig. 2)[6,7]. Overall, in triple-vaccinated individuals T cell responses remained robust. Activation of CD4 T cells showed a mild, yet significant increase in triple vaccinated individuals (Fig. 1a), while there was no difference in mean activation of CD8 T cells between double and triple-vaccinated study

participants (Fig. 1b). These results suggest that the boost delivered by the third vaccination primarily affected CD4 T cell reactivity. However, it is notable that the CD8 response in three individuals was particularly strong, suggesting that subsets of individuals develop a strong CD8 reaction after the third vaccination. Furthermore, we detected a small, yet significant decrease in CD4 T cell reactivity to the Prot_S B.1.1.529 Mutation Pool, as compared to the reference pool in double-vaccinated individuals (Fig. 1a). In contrast, CD8 T cell responses were not affected by B.1.1.529 mutations (Fig. 1b). These experimental results are supported by a previous study reporting that more of the known CD4 epitopes (~28%) than CD8 epitopes (14%) are altered by B.1.1.529 mutations[5]. Hence, a CD4 response would be expected to be more likely affected by such mutations. Despite the low number of study subjects, the statistical power of the analyzed CD4 T cell frequencies was 0.8 (Supplementary Fig. 3).

To evaluate the extent to which the T cell response to Spike protein originates from those sequences of the Spike protein, which contain a position which is mutated, the ratio of the T cell responses triggered by the Prot_S WT Reference pool to the Prot_S Complete pool was calculated. We determined that that approximately 56% and 35% of the total CD4 T cell response in double- and triple-vaccinated study subjects, respectively, correlated to regions containing the B.1.1.529 mutations (Fig. 1c). A similar correlation was found for approximately 83% and 71% of the total CD8 response in double- and triple-vaccinated study subjects, respectively (Fig. 1d). These percentages may strongly vary in vivo between individuals, since presentation of a mutated peptide is dependent on individual MHCs and processing of the complete antigen. In addition, the sum of reactivities of single peptide stimulation does not necessarily equal the stimulation with a pool of all peptides, thereby potentially leading to an overestimation of the proportion of the mutation-site restricted T cell response in vitro. Although the relative values might be to a certain degree different in vivo, for both CD4 and CD8 T cells, the proportion of the T cell response triggered by regions affected by B.1.1.529 mutations diminished following the third vaccination, suggesting a proportion of T cells were activated via unmutated sequences within the vaccines. These findings reveal a limited impact of B.1.1.529-related mutations on the memory T cell response, in agreement with recent reports[5,8].

To investigate altered MHC presentation to most widely expressed MHC-alleles in the European-Caucasian population, and to support the in vitro approach, all 222 possible 9-mer peptides (potentially binding to MHC class I molecules) and 83 15-mer peptides (that potentially bind to MHC class II molecules) containing B.1.1.529 mutations were analyzed in silico (Supplementary Fig. 4). Using NetMHCpan and NetMHCIIpan, changes in MHC binding were calculated in order to predict B.1.1.529 specific conversions amongst no-binding (NB), weak-binding (WB) and strong-binding (SB) peptides. This approach predicted for the majority of the most common DRB-1 alleles, conversions of up to 4 WB into NB peptides (Supplementary Data 1, Fig. 2a), and for the most common MHC-I alleles, conversion of up to 9 WB and SB into NB peptides (Supplementary Data 1, Fig. 2b). Based on the small number of conversions into NB peptides, certain individuals with a specific MHC background might have a disadvantage in immune recognition and potentially a small increase in the risk of developing a chronic or residual infection. However, a possible loss of an epitope for a specific MHC-allele, due to a mutation in the peptide sequence, might be compensated by one or more of the other MHC-alleles expressed by the individual, thereby leading to a net change of zero in the T cell response. Hence, based on the in silico calculations alone a disadvantage for a specific MHC cannot be proven. In our calculations, conversions were most prominent within the MHC-I

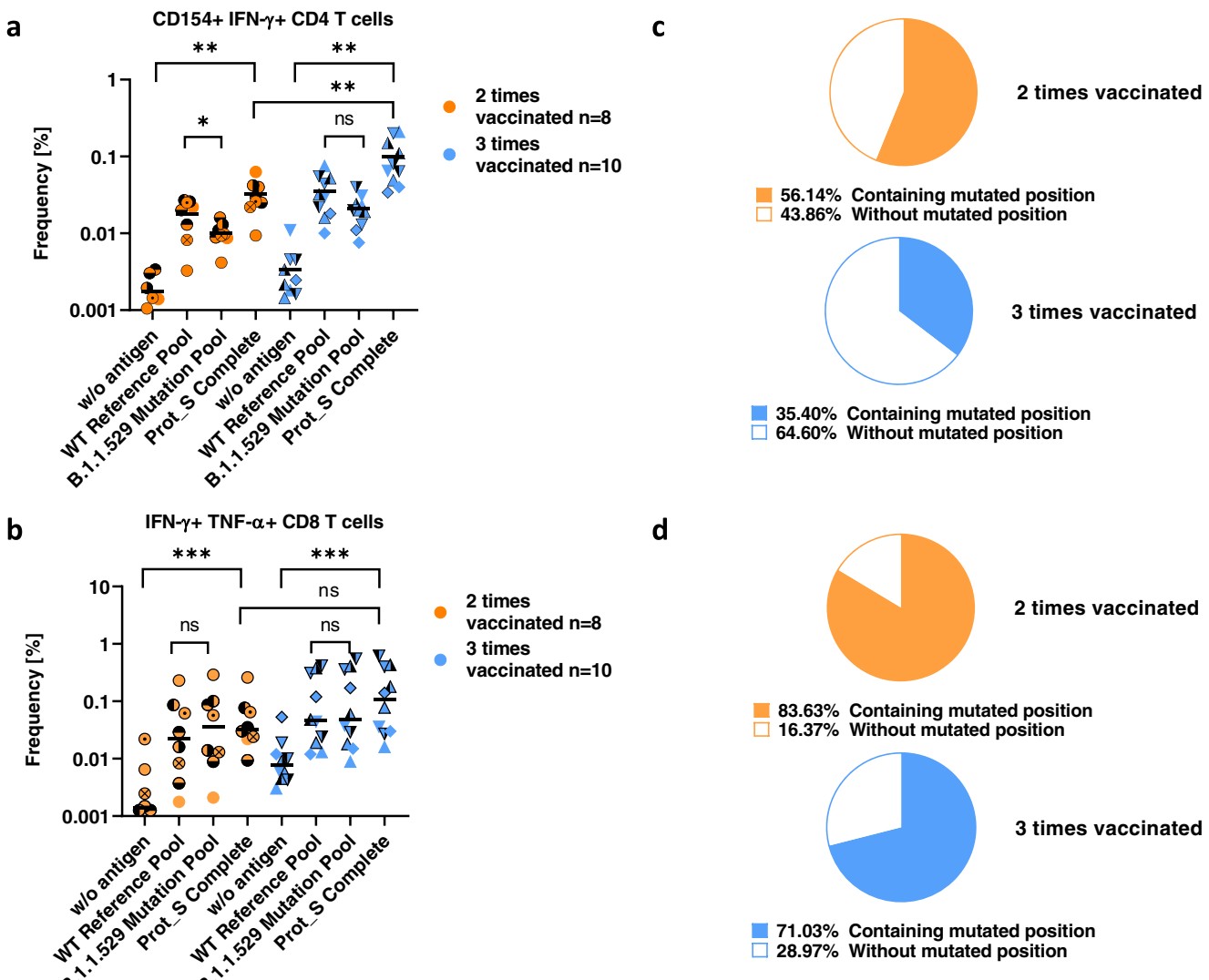

**Fig. 1 T Cell reactivity in twice and triple vaccinated individuals upon stimulation with ancestral and B.1.1.529 Spike protein derived peptides. a** The frequency [%] of CD154 + IFN-γ + cells among CD4 T cells and (**b**) of IFN-γ+TNF-α+ cells among CD8 T cells are shown on the *Y*-axes. Horizontal lines indicate the mean. The *X*-axes indicate the peptide pools used for stimulation. Each symbol represents a single individual. Orange refers to 2 times ($n = 8$) and blue to 3 times ($n = 10$) vaccinated individuals. As CD4 T cells are normally distributed, student´s T tests with Welch correction were used for calculation of significance. As CD8 T cells are not normally distributed, Mann–Whitney U-tests were performed. n.s. not significant, $*p < 0.05$, $**p < 0.01$, $***p < 0.001$. **c** Cake diagrams to determine the amount of the CD4 response and (**d**) the CD8 response originating from the unmutated peptides, which contain a mutated amino acid in B.1.1.529. The color-filled portions indicate the amount of the T cell response from the wildtype peptides, which contain a mutation in B.1.1.529. Empty areas refer to the T cell response from peptides without mutations. Orange refers to 2 and blue to 3 times vaccinated individuals.

group, with up to 9 peptides potentially no longer presented via HLA-B*14:02 (Fig. 2b). Of note, HLA-B*14:02 was associated with individuals showing viremic control of HIV[9], and a person living with HIV (PLHIV) having a residual SARS-CoV-2 infection was suggested as the origin of B.1.1.529[10]. However, nine peptides being affected might represent only a very small proportion of the total MHC class I epitopes presented via potentially six different MHC class I alleles, and might not be sufficient to cause a residual SARS-CoV-2 infection. Therefore, experimental validation in a selected larger study group would be required to confirm that HLA-B*14:02 might be associated with impaired T cell activation and chronic SARS-CoV-2 infection or residual Covid-19 disease in PLHIV. In general, in silico evaluations of changes in peptide binding affinities to most widely expressed MHC-alleles allows certainly for a first estimation on the effects of B.1.1.529-related mutations. Ultimately, in silico predictions

require in vitro verification assays for specific epitopes and their corresponding MHCs to prove the immunological consequences of algorithm based predictions.

In summary, based upon the frequencies of T cell responses, only a limited proportion of vaccine-induced SARS-CoV-2 reactive T cells are not capable to respond to peptides harboring B.1.1.529 mutations. These findings were recently corroborated by Geurts van Kessel et al. comparing T cell responses upon vaccination with different vaccines showing a minimal escape of B.1.1.529 at the T cell level[11]. As B.1.1.529 infections result in milder disease courses, although existing antibodies are less efficient in neutralizing the virus, T cells may confer protection from more severe disease courses to a currently unknown extent. Hence, further studies will need to address the impact of SARS-CoV-2 reactive T cells on protective immunity, in order to benefit future vaccine developments and vaccination campaigns.

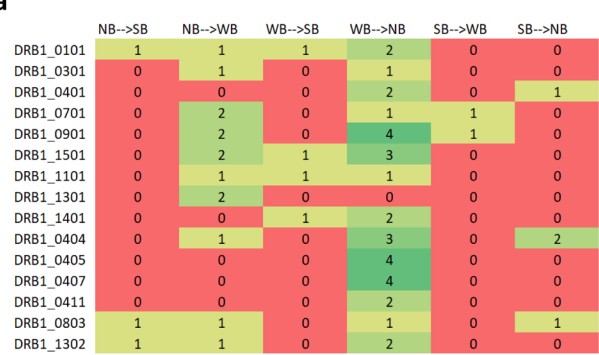

**Fig. 2 In silico prediction and comparison of strong-, weak- and no-binding peptides derived from the ancestral Wuhan and B.1.1.529 Spike proteins.**
**a** The *X*-axis indicates the potential conversions of non-binding (NB), weak binding (WB) or strong binding (SB) 9-mer peptides and the *Y*-axis indicates the most common DRB1 alleles in the European-Caucasian population. **b** Corresponding analysis to (**a**) highlighting the most common HLA-I alleles in the European-Caucasian population on the *Y*-axis. The numbers and the color code indicate the number of 9-mer peptides with predicted in silico conversions in B.1.1.529 compared to the ancestral Wuhan strain for the respective HLA allotype. Red: low conversion rate; Green: high conversion rate.

## Data availability

The data generated during and/or analyzed during the current study are available from the corresponding author on reasonable request and require permission of the study participants. Supplementary Data 1 contains the data underlying Figs. 1 and 2.

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

## Acknowledgements

This project was financially supported by the State of North Rhine-Westphalia, Germany, as part of the research grant "SARS-CoV-2 specific T-cell diagnostic", project no. MIL-1-1. We thank all participants of the study for blood donation.

## Author contributions

C.D., V.T. and N.J. performed the experiments. J.K., H.B., G.L. and Y.H. were responsible for blood donation and the ethical approval. S.K. and O.B. designed and provided the peptide pools. C.D., H.W., J.S., C.Do, A.R. and M.S. oversaw and conceived the study. C.D. and M.S. wrote the manuscript.

## Competing interests

C.D., V.T., N.J., J.K., H.B., G.L., Y.H., S.K., O.B., J.S., C.Do, A.R. and M.S. are employees of Miltenyi Biotec. No competing interest exists for the remaining author, H.W.
