## [Peer Review File · Communications Medicine]

Reviewers' comments:

Reviewer #1 (Remarks to the Author):

In this manuscript, the authors study the T cell responses against the Omicron variant in individuals vaccinated with 2 and with 3 doses. Their results support multiple studies that also demonstrated a robust T cell response against the Omicron variant. The analysis they provide is new as they explore the effects at the epitope level, which has not been explored by many studies. I find the paper interesting, concise, and well-written but I have a few major comments for the authors to address:

- Why is “without antigen” always greater in 3rd dose data in Fig. 1A and 1B? Is there some inherent bias in the results?
- n is too small. While results seem to be statistically significant, the significance seem to be low specifically for CD4 responses. I wonder if it is possible to increase n.
- When was the sampling done? How many weeks/months after 2nd and 3rd dose? Are the high titers after 3rd dose due to sampling done early after vaccination? This is important to clarify.
- While the result of Fig. 1C is interesting, it may be possible that the results are biased due to the HLA types of the individuals. Less mutated epitopes in 3 dose individuals but this might be due to their specific HLAs. Also, it would be good to clarify somewhere that the samples are not matched.
- Was there a difference in T cell response among individuals who got 3rd dose Comirnaty and those who got Spikevax? Is there any difference in results in Fig. 1 if the analysis considers the 3rd dose vaccine type?
- “it’s reasonable to assume that only persons with a specific MHC background might have a certain disadvantage in immune recognition, which might potentially lead to a chronic or residual infection in these individuals.” This statement is a bit misleading as 9 epitopes being affected may still be a very small number out of all possible epitopes that are targeted by HLA-B*07:02. Also a person has 6 HLA-I alleles. Thus, epitopes of one HLA in a patient being affected does not imply that T cell response of that person would be affected. This has to be made clear.
- 71-73: I don’t understand why this is interesting. How are B*14:02 results interesting in any way based on the results?
- Because of the above issues with Fig. 2 and the related text, it seems to be a weak part of the paper. I would recommend authors to focus only on the first part which is very interesting, and possibly remove the in Silico analysis. They can expand the analysis in Fig. 1 along different directions, which I mentioned above.

[EDITOR COMMENT: Given the other reviewer comments I would retain figure 2.]

Reviewer #2 (Remarks to the Author):

Sobczak et. al., performed a small flow-based analysis of twice- and triple-vaccinated individuals. The study does not add to current literature per se; they are consistent with previous studies demonstrating that the Omicron B.1.1.529 variant leads to diminished memory T cell responses to vaccination but that the impact is relatively small. In addition, the authors performed an in silico analysis to determine whether the 222 possible 9-mers studied in their experiments, predicted altered MHC presentation (and subsequent functional immune recognition). These in silico analyses suggested that HLA-B*14:02 individuals may possibly have poorer immune recognition (following

vaccination), in the setting of B.1.1.529. The authors suggest that there may be a potential link with HIV in that this HLA type has been associated with env-specific CD8+ T cell antiviral activity in HIV-1 disease.

Overall, the study is small and harbors several weaknesses. The most prominent weakness is that it is totally unclear if the individuals in the study were ever previously infected with SARS-CoV-2 infection, as non-spike responses were not analyzed or clinical history of prior COVID-19 infection is not described. Secondly, even the in silico analysis is limited by power and direct assays to determine immune recognition would have enhanced the study's findings. Finally, as a more minor comment, the short paper would benefit from some additional editing for English grammar (e.g., line 28: "However, these mutations do apparently not strongly abrogate MHC binding or restriction . . ." and line 74: ". . . only a limited proportion of vaccine induced SARS-CoV-2 reactive T cells are not capable to respond to peptides . . .").

Reviewer #3 (Remarks to the Author):

Sobczak and co-workers at the Miltenyi Biotec laboratory analyzed the cross-reactivity of CD4+ and CD8+ T cells induced by 2 or 3 doses of COVID-19 mRNA vaccination against the omicron VOC. Overall, the study is straight forward and confirms other very recent studies (mostly published as preprints, a small number also as peer reviewed publications) that demonstrate a high degree of cross-recognition.

There are, however, some important limitations of the methodological approach selected by the authors:

1. The authors use a complete set of 360 overlapping peptide (olp) set covering the ancestral spike protein (Prot_S Complete), but only a subset of 83 olps that contain the omicron mutations (Prot_S B.1.1.529 Mutation Pool) that is compared to the corresponding 83 ancestral olps (Prot_S WT Reference Pool). This is problematic for several reasons:
 - a) The 360 olps of Prot_S Complete cover the 1273 aa of Spike. Theoretically, 83 olps in the subsets could cover up to 1245 aa of spike (83 x 15). How many aa do they cover indeed? This is an important information, e.g. to interpret the data presented in Fig. 1C+D!
 - b) The response against a large number of olps does not equal the sum of the responses against the first half + the second half. Therefore, the following calculation made in Fig. 1C+D is not valid:
response against non-mutated olps = response against total olps – response against mutated olps
 - c) In addition, a response against a 15-mer olp that contains a mutation does not indicate that the targeted T-cell epitope (9-mers for CD8+ !) are mutated, also limiting the validity of Fig. 1C+D. These limitations need to be discussed openly, and Fig. 1C+D should be discussed much more carefully. To overcome these limitations of the olp sets that are currently available at Miltenyi (according to their homepage), I would also recommend that Miltenyi establishes a Pro_S_Complete B.1.1.529 set.
2. In the MHC binding prediction analysis (Fig. 2), the authors test 9-mers to predict binding to both MHC-I and MHC-II. This is suitable for MHC-I, since CD8+ T cell epitopes are usually 9-mers, it is definitively not suitable for MHC-II, since CD4+ T cell epitopes are longer (15-mers, with more variation compared to CD8+ epitopes). These analyses for MHC-II should be repeated with the

suitable settings (15-mers, as default setting in netMCHIIpan).

Minor:

3. The authors need to clarify that only BA.1 subtype of omicron was studied (at the Miltenyi homepage, BA.2 olps are also available).
4. The peptide sets should be introduced with more details: overlapping peptides, with 15-mers overlapping by 11 aa.
5. Introductory paragraph: "Both CD4 and CD8 responses... were marginally reduced... corresponding to MHC binding analysis": MHC binding analysis cannot measure T cell responses. Please rephrase e.g. to "MHC binding analysis suggests that both CD4 and CD8 responses may be marginally reduced...".
6. Line 61: "all possible 9-mer core peptides..." What is referred to as "core"?
7. Line 71: Correct "HLA-B14:02" to "HLA-B*14:02" (2x).
8. Suppl. Table 1: Please delete the exact dates of 1st, 2nd and 3rd vaccination, since these exact dates probably would allow extraction of the individual donors from national vaccination databases, not complying with data safety regulations. Please instead indicate the days post first vaccination, and also the relative date of sampling, which is currently lacking (so e.g. d0 first vaccination, d28 second vaccination, d210 third vaccination, d250 blood sampling).
9. Fig. 1A+B: I am puzzled by the statistics. I only the complete olp pool (**), but not the two smaller olp pools significant versus negative control? In addition, the range of the y axis in A versus B should be the same, since there are now data points >1% in panel B.
10. Fig. 2C+B: I do not understand the order of the rows. It would make sense to display the conditions with increasing binding e.g. on the left and conditions with decreasing binding e.g. on the right. In addition, I find red for low numbers quite contra-intuitive.

Current SARS-CoV2 vaccines provide T cell memory to the B.1.1.529 variant (COMMSMED-22-0071-T) – Point-by-point reply to Nature Communication Medicine

Charlyn Sobczak¹, Verena Traska¹, Nicole Jansen¹, Julia Kostyra¹, Herrad Baurmann¹, Gereon Lauer¹, Yi-Ju Huang¹, Sven Kramer¹, Olaf Brauns¹, Holger Winkels², Jürgen Schmitz¹, Christian Dose¹, Anne Richter¹ and Marc Schuster^{1,3}

This point-by-point reply is written as part of the reviewing process of the above mentioned manuscript that has been submitted to the Journal *Nature Communication Medicine* under the manuscript ID COMMSMED-22-0071-T. We were thankful to incorporate the extensive and constructive scientific feedback given by all three reviewers, which improved the quality of the manuscript. In the following we will provide detailed answers to each of the reviewers' remarks, referring to our revised version of the manuscript.

Remarks by #Reviewer 1

1. *“Why is “without antigen” always greater in 3rd dose data in Fig. 1A and 1B? Is there some inherent bias in the results?”*

Reviewer 1 highlights the increased frequency of activated CD4 and CD8 T cells in the unstimulated samples, after the 3rd vaccination compared to individuals receiving only two doses. We are convinced that this is not an inherent bias in the results, as similar effects have been described before. Elevated expression of activation markers after (re-)challenging the immune system can lead to a minor activation of bystander T cells upon vaccination (Peixoto et al., 2007), suggesting that vaccination could have increased background levels of activated T cells in our experiments. To further corroborate our data, we provide additional in-house generated data tracking the levels of activated T cells in 29 individuals pre and post vaccination. Inhere we also found, a slight increase of activated CD4+ and CD8+ T cells in the unstimulated samples 2-3 weeks after vaccinations compared to pre-vaccination samples (see RL-Fig. 1). In summary, elevated levels of activated T cells in the unstimulated samples after the 3rd vaccination as we report here are most likely the result of SARS-CoV-2 vaccination. Whether the strength of mRNA vaccines in general causes increased bystander T cell activation, or whether this is a phenomenon associated to the selected antigen cannot be clarified in this study.

2. *“n is too small. While results seem to be statistically significant, the significance seem to be low specifically for CD4 responses. I wonder if it is possible to increase n”*

We thank the reviewer for the suggestion to improve the statistical significances of our analyses. Unfortunately, the number of n cannot be increased anymore, as all participants received the third vaccine dose by now and additional volunteers could not be recruited into the study group. Hence, we cannot improve the statistical significance of the results. To attenuate the reviewers concerns as to the meaning of our analysis for the CD4 response, we performed a power analysis to estimate the probability at which we detect differences within and/or between our cohorts and simultaneously prevent false positive and false negative results. The analysis, shown in RL-Figure 2, revealed at the chosen sample size of n= 8, for two times vaccinated individuals, and n=10 for three times vaccinated individuals a statistical power of 0.8 within the CD4+ T cell subset. Hence, an acceptable power for CD4+ T cell analysis was reached, albeit having only low significance calculated.

3. “When was the sampling done? How many weeks/months after 2nd and 3rd dose? Are the high titers after 3rd dose due to sampling done early after vaccination? This is important to clarify.”

We added the required information to increase the transparency and informational value of our study group as the reviewer proposed. Hence, we extended the supplementary table 1 and added the interim days that passed after the 2nd (mean 182 days / 26 weeks) and 3rd vaccination (mean 42 days / 5.9 weeks) until sample collection. To address the question whether high titers were found in individuals having recently received the 3rd vaccination we did a correlation analysis. The analysis revealed no relevant correlation between the interim days, the day of third vaccination and the day of sample collection, and the reactivity of neither CD4+CD154+IFN γ + T cells nor CD8+TNF α +IFN γ + T cells (See RL-Figure 3).

4. “While the result of Fig. 1C is interesting, it may be possible that the results are biased due to the HLA types of the individuals. Less mutated epitopes in 3 dose individuals but this might be due to their specific HLAs. Also, it would be good to clarify somewhere that the samples are not matched.”

We thank the reviewer for this comment. To address this point, we now point out in the manuscript that samples are not matched (see line 36-39) and addressed this concern by mentioning the impact that non-matched HLA-alleles between the two different cohorts might have on the analyses (see line 37-39), in the manuscript. To further bolster that point and to increase transparency, we also added in this point by point reply an overview about the expressed HLA subtypes (RL-Figure 5).

5. “Was there a difference in T cell response among individuals who got 3rd dose Comirnaty and those who got Spikevax?” and “Is there any difference in results in Fig. 1 if the analysis considers the 3rd dose vaccine type?”

The reviewer draws attention to an interesting and important point which might bear potential novel insights as to the comparability of the different vaccines. Therefore, we performed the requested analysis (RL-Figure 6). It is important to mention that due to small sample numbers within the subgroups of three times vaccinated individuals (n=3 for three times vaccinated with Comirnaty (also called BNT162b2); n=6 for three times vaccinated with Spikevax (also called mRNA-1273)) the power of all performed tests is rather low. Hence, we decided to not split our analysis based on the vaccine type of the third dose for the data shown in the manuscript. However, since the reviewer highlights the importance of comparing vaccine effectiveness on the T cell level, we decided to include in the discussion a section referring to a recent study published by Corine Geurts van Kessel and her group comparing different SARS-CoV-2 vaccines on the T cells level (see line 98-99). In here, the authors reported a vaccine-dependent impact of the omicron-strain on the CD4+ T cell response shortly after (28-56 days) vaccination as well as 6 month post vaccination. The greatest reduction in CD4+ T cell response against Omicron-derived peptides was detected for individuals vaccinated with BNT162b2 vaccine (5-fold decrease compared to response against the Wildtype strain), followed by ChAdOX-1 S -, Ad26.COVS.2.S - and mRNA-1273 - vaccination, respectively.

6. “It’s reasonable to assume that only persons with a specific MHC background might have a certain disadvantage in immune recognition, which might potentially lead to a chronic or residual infection in these individuals.” This statement is a bit misleading as 9 epitopes being affected may still be a very small number out of all possible epitopes that are targeted by HLA-B0702. Also a person has 6 HLA-I alleles. Thus, epitopes of one HLA in a patient being affected does not imply that T cell response of that person would be affected. This has to be made clear“
7. “line 71-73: I don’t understand why this is interesting. How are B1402 results interesting in any way based on the results?”

We decided to formulate one response to both comments of the reviewer. We completely agree with the reviewer's remark that the broad variety of different HLAs and the fact that every person can express up to six different HLA class I alleles decreases the chance that the T cell response is affected in case only nine potential MHC class I epitopes show impaired binding. Therefore we adapted the discussion according to the reviewers comment (see line 88-92). Furthermore, we now point out that the effect of the mutations on the T cell responses towards these nine peptides would require further *in vitro* experiments in a larger study group. In addition, based on the reviewer's comment, we included a more detailed discussion about a potential association between HLA-B*14:02, PLHIV and residual disease and improved this section in the manuscript. Please see the respective changes in the discussion concerning HLA-B*14:02 and PLHIV in the revised manuscript (line 88-96).

8. "Because of the above issues with Fig. 2 and the related text, it seems to be a weak part of the paper. I would recommend authors to focus only on the first part which is very interesting, and possibly remove the *in Silico* analysis. They can expand the analysis in Fig. 1 along different directions, which I mentioned above."

[EDITOR COMMENT: Given the other reviewer comments I would retain figure 2.]

We thank the reviewer for the suggestion, but as instructed by the editor we kept figure 2 within the manuscript. However, since more detailed explanation seemed to be required for the *in silico* analyses of our work (which has been suggested by the other two reviewers as well), we adapted the manuscript accordingly and modified the supplementary figures 1 and 2, explaining our workflow schematically in more detail.

Remarks by #Reviewer 2

1. "Overall, the study is small and harbors several weaknesses. The most prominent weakness is that it is totally unclear if the individuals in the study were ever previously infected with SARS-CoV-2 infection, as non-spike responses were not analyzed or clinical history of prior COVID-19 infection is not described"

We thank the reviewer for the recommendation to include Covid-19 disease history of the recruited blood donors. Accordingly, we included that information in the manuscript showing that only one of the blood donors suffered from a documented prior natural infection with SARS-CoV-2 (line 30-33). We included that information in the supplementary table of the manuscript, which provides detailed information on the individual background of the blood donors.

2. "Secondly, even the *in silico* analysis is limited by power and direct assays to determine immune recognition would have enhanced the study's findings"

We fully agree with the reviewer as to the limitation of our *in silico* approach and its ability to characterise the impact specific T cell epitopes might have. We totally agree on this point as being the major limitation of *in silico* approaches predicting T cell activation, and therefore thank the reviewer for his/her remark. We took this as an opportunity to discuss such limitations within the revised manuscript (line 92-96).

3. Finally, as a more minor comment, the short paper would benefit from some additional editing for English grammar (e.g., line 28: "However, these mutations do apparently not strongly abrogate MHC binding or restriction . . ." and line 74: ". . . only a limited proportion of vaccine induced SARS-CoV-2 reactive T cells are not capable to respond to peptides . . .")."

We thank the reviewer for this remark. Prior to resubmission, the revised manuscript was carefully read through by a native English speaker to improve the quality of the manuscript.

Remarks by #Reviewer 3

1) “The authors use a complete set of 360 overlapping peptide (olp) set covering the ancestral spike protein (Prot_S Complete), but only a subset of 83 olps that contain the omicron mutations (Prot_S B.1.1.529 Mutation Pool) that is compared to the corresponding 83 ancestral olps (Prot_S WT Reference Pool). This is problematic for several reasons”

a) “The 360 olps of Prot_S Complete cover the 1273 aa of Spike. Theoretically, 83 olps in the subsets could cover up to 1245 aa of spike (83 x 15). How many aa do they cover indeed? This is an important information, e.g. to interpret the data presented in Fig. 1C+D!”

We thank the reviewer for raising this point, as it shows that more information about the design of our peptide pools is required. As mentioned by the reviewer, the peptides used within our *in vitro* stimulation approach are overlapping. The overlap of these 15mer consecutive peptides is in most cases 11 amino acids to each other. Consequently, our Prot_S Complete peptide pool, which comprises of in total 360 peptides, covers the complete spike protein sequence. Importantly, the same design (15mers with 11mer overlaps) is used within the smaller peptide pools and, hence, Omicron mutations are covered by multiple 15mer peptides, having the mentioned 11mer overlap. Therefore, 39.67% of the complete spike protein sequence is covered by the Prot_S B.1.1.529 Mutation Pool and the Prot_S WT Reference Pool.

Please note that the integration of 11mer overlaps between the consecutive peptides bears the advantage that all peptides that were selected for the smaller, mutational peptide pools comprises of a mutated position either at the mid part, the N- or the C-terminal part of the peptide. This ensures a proper peptide processing within our *in vitro* approach, as peptide processing and presentation by antigen presenting cells is not influenced by the prior selection of the position of the mutated amino acid within the presented peptide. This becomes especially important for MHC class I restricted peptides which typically are made up of ~9 amino acids.

As already mentioned by the reviewer, these details on the peptide design are important to be clarified for a correct interpretation of the data presented within our manuscript. To prevent misunderstandings, we added two supplementary figures 1 and 2 within our resubmitted manuscript, explaining the peptide design as well as the workflow of our *in vitro* and *in silico* approach in more detail.

b) “The response against a large number of olps does not equal the sum of the responses against the first half + the second half. Therefore, the following calculation made in Fig. 1C+D is not valid: response against non-mutated olps = response against total olps – response against mutated olps”

The reviewer brings up an important point concerning the calculation of the relative reactivity of the mutated regions to the total Spike protein. We fully agree with the reviewer that the sum of reactivities of single peptide stimulation does not equal the stimulation with a pool of all peptides. Biologically, this is most likely the results of weaker epitopes which can be presented in the absence of stronger binding competitor epitopes. We would like to point out that the main conclusion of this analysis as we present in the manuscript is the change from the second to the third vaccination. “...,for both CD4 and CD8 T cells, the proportion of the T cell response triggered by regions affected by B.1.1.529 mutations diminished following the third vaccination”. Following the reviewers comments, we did with groups of 4 individuals restimulation experiments using two separate pools of the Spike protein

(mutated peptides + total spike w/mutated peptides) and compared the sum of the pools with the stimulation of all peptides. As expected the stimulation with all peptides induced a lower T cell response, compared to the sum of the responses when the T cells were stimulated with the two pools independently. For CD4 T cells reactivity was about 44% higher and for CD8 T cells about 34% (RL-Fig.7A -C). As expected, when the results shown in Fig. 1C + 1D was corrected using these values there was a change in the relative values (RL-Fig. 7 D and E). However, our main conclusion that the “[...] proportion of the T cell response triggered by regions affected by B.1.1.529 mutations diminished following the third vaccination” was not affected. In summary, due to the reviewer’s comments we felt the need to include a section in the manuscript pointing out the uncertainty of the presented relative values due to the difference between single peptide and pooled stimulations, but kept our main conclusion as to the difference between the second and third vaccination (line 61-65), since this statement is not challenged by the reanalysis presented in this point-by-point reply.

- c) “In addition, a response against a 15-mer oIp that contains a mutation does not indicate that the targeted T-cell epitope (9-mers for CD8+ !) are mutated, also limiting the validity of Fig. 1C+D”

We thank the reviewer for this comment. Also here, we are in full agreement with the reviewer, since a mutation provides only a mutated epitope in case the test subjects MHCs are able to present the corresponding peptide derived from the mutated region. Furthermore, not all 9-mers, which are potentially derived from 15-mers due to uptake and intracellular processing and presentation or originate from dissociation in the media are generated in the experiment with the same probability. However, it’s fair to say that all *in vitro* antigen-specific T cell assays employing peptide pools for restimulation have this limitation, but are nonetheless, standard in antigen specific T cell analytics. Corresponding to our reply for point 1.b) we refer to the same section in the manuscript discussing this limitation (line 61-63) and in addition to the sections added in response to reviewers #1 (line 80-84 and line 88-90).

- 2) “In the MHC binding prediction analysis (Fig. 2), the authors test 9-mers to predict binding to both MHC-I and MHC-II. This is suitable for MHC-I, since CD8+ T cell epitopes are usually 9-mers, it is definitively not suitable for MHC-II, since CD4+ T cell epitopes are longer (15-mers, with more variation compared to CD8+ epitopes). These analyses for MHC-II should be repeated with the suitable settings (15-mers, as default setting in netMHCIIpan).”

We thank the reviewer for this remark. We want to point out that we applied the NetMHCIIpan algorithm with the sequence or 15-mer peptides, however, our description in the manuscript was misleading. Hence, we modified the schematic overview shown in supplementary figure 2, explaining the workflow of our *in silico* approach in more detail. We hope that we are explaining the strategy more clearly now in the revised manuscript.

3) Minor comments:

We highly appreciate the extended and detailed minor feedback points given by the third reviewer. We took care of each point of the reviewer and improved the manuscript at the required points as followed:

- The authors need to clarify that only BA.1 subtype of omicron was studied (at the Miltenyi homepage, BA.2 oIps are also available).

Please see line 34 in the revised manuscript.

- The peptide sets should be introduced with more details: overlapping peptides, with 15-mers overlapping by 11 aa.

Please see suppl. Figure 1 in the revised manuscript.

- Introductory paragraph: “Both CD4 and CD8 responses... were marginally reduced... corresponding to MHC binding analysis”: MHC binding analysis cannot measure T cell responses. Please rephrase e.g. to “MHC binding analysis suggests that both CD4 and CD8 responses may be marginally reduced...”.

Please see line 15 in the revised manuscript.

- Line 61: “all possible 9-mer core peptides...” What is referred to as “core”?

The term core peptide refers to the minimal number of amino acids required to bind to a certain MHC.

- Line 71: Correct “HLA-B14:02” to “HLA-B*14:02” (2x).

In the entire manuscript “HLA-B*14:02” is now used.

- Suppl. Table 1: Please delete the exact dates of 1st, 2nd and 3rd vaccination, since these exact dates probably would allow extraction of the individual donors from national vaccination databases, not complying with data safety regulations. Please instead indicate the days post first vaccination, and also the relative date of sampling, which is currently lacking (so e.g. d0 first vaccination, d28 second vaccination, d210 third vaccination, d250 blood sampling).

Exact dates were deleted.

- Fig. 1A+B: I am puzzled by the statistics. Is only the complete olp pool (**), but not the two smaller olp pools significant versus negative control? In addition, the range of the y axis in A versus B should be the same, since there are now data points >1% in panel B.

Regarding the last point, referring to our statistics, we would like to clarify all indicated significances:

Looking into the analysis of CD4+CD154+IFN γ + T cells, presented in Fig. 1A we found a statistically significant increase ($p < 0.01$) between the native controls (w/o antigen) and cells being stimulated with the Prot_S Complete, within both cohorts. The same holds true when comparing the frequency of activated T cells upon stimulation with Prot_S Complete peptides going from 2x to 3x vaccinated individuals. A slightly smaller, but still significant difference ($p < 0.05$) was found within the group of 2x vaccinated individuals, between the two corresponding peptides pools, WT Reference and B.1.1.529. This significance was neither observed for 3x vaccinated individuals, nor for both cohorts when looking into CD8+IFN γ +TNF α + T cells (Fig. 1B). Within the latter, also the inter-cohort comparison did not reveal significant changes between cells being stimulated with the Prot_S Complete peptide pool. Here, we could solely detect, in alignment with the CD4+ T cell compartment, a significant increase in activated cell frequencies comparing the negative controls with cells being stimulated with Prot_S Complete peptides ($p < 0.001$).

RL-Figure 1 Frequencies of CD4+ CD154+ TNF- α + and CD8+ TNF- α + IFN- γ + T cells w/o with peptide stimulation

The frequency [%] of (A) CD4+ CD154+ TNF- α + and (B) CD8+ TNF- α + IFN- γ + T cells (y-axis) is plotted for the three measurements: pre (black dots), post 1st(orange dots), and post 2nd vaccination (blue dots)(x-axis) with Covid-19 vaccines. Error bars show the standard deviation (SD) with the mean. Data acquisition was performed at 23 days post 1st and 15 days post 2nd vaccination, on average (n=29).

Statistical power calculation curve

RL-Figure 2 Statistical power analysis for the in vitro testing of CD4+ T cell responses towards antigen stimulation.

Given is the power under the current number of n ($n=10$ for 3x vaccinated individuals and $n=8$ for 2x vaccinated individuals). The significance threshold α is set to 0.05 for all calculations. As standard deviation (SD) the greatest SD found within the subgroups of CD4+ T cells stimulated with the three different peptide pools was set for the model calculation.

RL-Figure 3 Correlation analysis of the frequencies for CD4+CD154+IFN- γ + T cells (y-axis, left plots) and CD8+TNF- α +IFN- γ + T cells (y-axis, right plots) detected upon stimulation with three different peptide pools with the interim days from 3rd vaccination until sample collection (x-axis). Linear trendlines with associated R^2 -values are added to each to plot.

RL-Figure 4 Changes in CD8+IFN- γ +TNF- α + T Cell reactivity upon stimulation with B.1.1.529 peptides with increased sample size (n) The frequency [%] of IFN- γ +TNF- α + cells among CD8 T cells are shown on the y-axes. Horizontal lines indicate the mean. The x-axes indicate the peptide pools used for stimulation. Each symbol represents a single individual. Orange refers to 2 times and blue to 3 times vaccinated individuals. As CD8 T cells are not normally distributed, Mann-Whitney U-tests were performed. n.s. = not significant, *** $p < 0.001$.

MHC class I allotypes

MHC class II allotypes

RL-Figure 5 Distribution of HLA class I (top row) and HLA class II (lower row) allotypes within the cohorts of 2x (orange) and 3x (blue) vaccinated individuals. Shown are the differential HLA class I and HLA class II allotypes (y-axis) plotted against the absolute number of donors sequenced with these alleles (x-axis). Non-classical HLA-codes bearing a letter-code represent multi-allele codes.

T cell responses upon booster with Spikevax / Comirnaty

RL-Figure 6 Changes in T Cell reactivity upon stimulation with B.1.1.529 peptides within 2x vaccinated individuals and 3x vaccinated individuals boosted with either Comirnaty (left) or Spikevax vaccine (right). The frequency [%] of CD154+IFN- γ + cells among CD4 T cells (lower graphs) and of IFN- γ +TNF- α + cells among CD8 T cells (upper graphs) are shown on the y-axes. Horizontal lines indicate the mean. The x-axes indicate the peptide pools used for stimulation. Each symbol represents a single individual. Orange refers to 2 times and blue to 3 times vaccinated individuals. As CD4 T cells are normally distributed, student's T tests with Welch correction were used for calculation of significance. As CD8 T cells are not normally distributed, Mann-Whitney U-tests were performed. n.s. = not significant, * $p < 0.05$, ** $p < 0.01$, *** $p < 0.001$.

RL-Figure 7 Correction of relative reactivities of mutated regions to the complete spike protein. Frequency of activated CD4+CD154+IFN γ + (A) and CD8+IFN γ +TNF α + T cells (B) upon stimulation with peptide pools covering the complete Omicron spike protein ("Omicron Complete S"), or comprising solely of the mutated regions ("Omicron Mutated Regions"), or solely of the respective non-mutated regions ("WT spike w/o mutated regions"). Table C comparably shows the frequency-values measured upon stimulation with the complete spike protein, as well as the calculated sum of the frequencies upon the stimulation with the two partial peptide pools. Furthermore, the ratio of these two values as well as the percentual difference is calculated, showing a mean difference of 44 % for CD4+ T cells and 34% for CD8+ T cells. Taking into account these differences, the corrected relative reactivities of mutated regions to the complete spike protein are corrected for both T cell subsets (D: CD4+ T cell subset; E: CD8+ T cell subset)

References

- Peixoto, A., Evaristo, C., Munitic, I., Monteiro, M., Charbit, A., Rocha, B., & Veiga-Fernandes, H. (2007). CD8 single-cell gene coexpression reveals three different effector types present at distinct phases of the immune response. *Journal of Experimental Medicine*, 204(5), 1193–1205. <https://doi.org/10.1084/jem.20062349>

REVIEWERS' COMMENTS:

Reviewer #1 (Remarks to the Author):

I would like to thank the authors for thoroughly addressing my comments. The revisions have greatly improved the work. I only have a couple of minor suggestions.

1. Regarding comment 2: Can the authors elaborate more on this test and how exactly they implemented it? What data exactly they used, what was considered true positive false positive etc. I find the current description vague.

And it would be good to include the associated figure as a supporting figure to show confidence in their results.

2. Regarding comment 3: Thanks for adding this important information. The average sampling for 2nd and 3rd dose is suggestive that the titres observed after the 3rd dose may be appearing higher than those after 2nd dose due to sampling being done earlier after 3rd dose. Based on these factors being different, I think the authors should pay more focus on the fact that the overall message of the work should be that the T cell response appears robust rather than claiming strongly about T cell response being “higher” after 3rd dose when compared with 2nd dose.

I am not sure the correlation analysis done addresses my question since none of the samples that you have after 3rd dose go beyond 100 days and in fact most of them are clustered around the mean of 40 days. Also, for addressing this specific question, a proper test would be to track the trend of T cell response within-host over time. Anyway, I think this is not a major issue since the authors are providing the sampling time now.

Reviewer #2 (Remarks to the Author):

Sozback et. al., addressed the three reviewer comments, which upon reading, appear to have overarching themes – which is that the data are not novel, the sample size is small, and although the epitope analysis is interesting, overall, the in silico analysis does not add much to the paper, and caution should be used in over-interpreting these data. While I appreciate the authors’ responses, my professional opinion is that this study is of limited value – and unfortunately, even less so now that Omicron BA.4 and BA.5 are circulating; we are well past BA.1, and it is clear that these newer variants, while mild, appear to behave similar to the BA.1, which was the only variant studied here. Nonetheless, the authors made some efforts to improve the English grammar, which is to be commended (although I would argue that the new highlighted sentences are still somewhat confusing, with double negatives, etc., and it is still unclear if the COVID-infected individual was also double- and triple-vaccinated, based on the way the sentence is written).

Reviewer #3 (Remarks to the Author):

The authors have addressed my previous points of concern. The respective experimental limitations

are stated/discussed in the revised manuscript.

Current SARS-CoV-2 vaccines provide T cell memory to the B.1.1.529 variant (COMMSMED-22-0071-T) – Point-by-point reply to Nature Communication Medicine

Charlyn Dörnte¹, Verena Traska¹, Nicole Jansen¹, Julia Kostyra¹, Herrad Baurmann¹, Gereon Lauer¹, Yi-Ju Huang¹, Sven Kramer¹, Olaf Brauns¹, Holger Winkels², Jürgen Schmitz¹, Christian Dose¹, Anne Richter¹ and Marc Schuster^{1,3}

¹Miltenyi Biotec B.V. & Co. KG, Friedrich-Ebert-Straße 68, 51429 Bergisch Gladbach, Germany

²University of Cologne, Faculty of Medicine and University Hospital Cologne, Clinic III for Internal Medicine, Cologne, Germany

³Corresponding Author: marcsch@miltenyi.com

We are thankful to incorporate the final feedback given by all three reviewers. In the following we will provide detailed answers to each of the reviewers' remarks, referring to our final submission of the manuscript.

Reviewer #1 (Remarks to the Author):

I would like to thank the authors for thoroughly addressing my comments. The revisions have greatly improved the work. I only have a couple of minor suggestions.

We thank the reviewer for the scientific input during the entire review process and have incorporated the final suggestions to improve the quality of the manuscript.

1. Regarding comment 2: Can the authors elaborate more on this test and how exactly they implemented it? What data exactly they used, what was considered true positive false positive etc. I find the current description vague. And it would be good to include the associated figure as a supporting figure to show confidence in their results.

We thank the reviewer for the suggestion. The associated figure and calculation, which have been part of the previous point-by-point reply was added to the final manuscript as Supplementary Figure 2b.

2. Regarding comment 3: Thanks for adding this important information. The average sampling for 2nd and 3rd dose is suggestive that the titres observed after the 3rd dose may be appearing higher than those after 2nd dose due to sampling being done earlier after 3rd dose. Based on these factors being different, I think the authors should pay more focus on the fact that the overall message of the work should be that the T cell response appears robust rather than claiming strongly about T cell response being "higher" after 3rd dose when compared with 2nd dose.

I am not sure the correlation analysis done addresses my question since none of the samples that you have after 3rd dose go beyond 100 days and in fact most of them are clustered around the mean of 40 days. Also, for addressing this specific question, a proper test would be to track the trend of T cell response within-host over time. Anyway, I think this is not a major issue since the authors are providing the sampling time now.

We thank the reviewer for the comment. We rephrased our conclusions in the manuscript. T cell responses are now indicated as "robust" where applicable.

Reviewer #2 (Remarks to the Author):

Sozback et. al., addressed the three reviewer comments, which upon reading, appear to have overarching themes – which is that the data are not novel, the sample size is small, and although the epitope analysis is interesting, overall, the in silico analysis does not add much to the paper, and caution should be used in over-interpreting these data. While I appreciate the authors' responses, my professional opinion is that this study is of limited value – and unfortunately, even less so now that Omicron BA.4 and BA.5 are circulating; we are well past BA.1, and it is clear that these newer variants, while mild, appear to behave similar to the BA.1, which was the only variant studied here. Nonetheless, the authors made some efforts to improve the English grammar, which is to be commended (although I would argue that the new highlighted sentences are still somewhat confusing, with double negatives, etc., and it is still unclear if the COVID-infected individual was also double- and triple-vaccinated, based on the way the sentence is written).

Reviewer #3 (Remarks to the Author):

The authors have addressed my previous points of concern. The respective experimental limitations are stated/discussed in the revised manuscript.

We thank both Reviewers #2 and #3 for the scientific input during the entire review process and are happy to have fully addressed the scientific comments as well as discussions regarding the limitations of the study and thereby improved the quality of the manuscript.